# Potential effect of tolvaptan on polycystic liver disease for patients with ADPKD meeting the Japanese criteria of tolvaptan use

Hiroki Mizuno[1,2]*, Akinari Sekine[3], Tatsuya Suwabe[1], Daisuke Ikuma[1], Masayuki Yamanouchi[1], Eiko Hasegawa[3], Naoki Sawa[1], Yoshifumi Ubara[1], Junichi Hoshino[2,3]

1 Nephrology Center, Toranomon Hospital Kajigaya, Kawasaki, Kanagawa, Japan, 2 Okinaka Memorial Institute for Medical Research, Tokyo, Japan, 3 Nephrology Center, Toranomon Hospital, Tokyo, Japan

* hilomiz@yahoo.co.jp

**Data Availability Statement:** All relevant data are within the manuscript and its Supporting Information files.

## Abstract

Polycystic liver disease (PLD) is a common extrarenal complication of autosomal dominant polycystic kidney disease (ADPKD), which causes compression-related syndrome and ultimately leads to liver dysfunction. Tolvaptan, a V2 receptor antagonist, is widely used to protect kidney function in ADPKD but its effect on PLD remains unknown. An observational cohort study was conducted to evaluate tolvaptan's effect on patients with PLD due to ADPKD. After screening 902 patients, we found the 107 ADPKD patients with PLD who met the criteria of tolvaptan use in Japan. Among them, tolvaptan was prescribed for 62 patients (tolvaptan group), while the other was defined as the non-tolvaptan group. Compared with the non-tolvaptan group, the tolvaptan group had larger height-adjusted total kidney volume (median 994(range 450–4152) mL/m, 513 (405–1928) mL/m, p = 0.01), lower albumin level (mean 3.9±SD 0.4 g/dL, 4.3±0.4g/dL, p<0.01), and higher serum creatinine level (1.2±0.4 mg/dL, 0.9±0.2 mg/dL, p<0.01). Although the median change in annual growth rate of total liver volume (TLV) was not statistically different between the tolvaptan group (-0.8 (-15.9, 16.7) %/year) and the non-tolvaptan group (1.7 (-15.6–18.7) %/year)(p = 0.52), 20 (43.5%) patients in the tolvaptan group experienced a decrease in the growth rate of TLV (responders). A multivariable logistic regression model adjusting for related variables showed that older age (odds ratio 1.15 [95% CI 1.01–1.32]) and a higher growth rate of TLV in the non-tolvaptan period (odds 1.45 95% CI 1.10–1.90) were significantly associated with responders. In conclusion, the change in annual growth rate of TLV in ADPKD patients taking tolvaptan was not statistically different compared with that in ADPKD patients without taking tolvaptan. However, tolvaptan may have the potential to suppress the growth rate of TLV in some PLD patients due to ADPKD, especially in older patients or those that are rapid progressors of PLD. Several limitations were included in this study, therefore well-designed prospective studies were required to confirm the effect of tolvaptan on PLD.

**Funding:** This study was supported by the grant from Otuska pharmaceutical company (IST-JPN-000230), however, this does not alter our adherence to PLOS ONE policies on sharing data and materials.

**Competing interests:** This work was supported by a commercial source, J.H's competitive research grant from Otsuka Pharm, Japan. This commercial funder, however, did not relate to employment, consultancy, patents, products in development, marketed product nor had influenced study design, data collection and analysis, decision to publish, or preparation of the manuscript. In addition, this funder does not alter our adherence to PLOS ONE policies on sharing data and materials.

## Introduction

Polycystic liver disease (PLD) is a rare inherited disease characterized by the development of multiple cysts in the liver [1, 2]. Although genetic analysis has revealed that various genes are related to PLD [3], more than ninety percent of older patients with autosomal dominant polycystic kidney disease (ADPKD) have hepatic cysts [4]. In severe cases, the mass effects reduce patient quality of life and sometimes lead to mortality due to liver dysfunction [5–7]. Previous studies have shown that the progression of PLD was faster in specific patients, especially in younger women, which suggested that estrogen was related to cyst growth [8, 9].

Current medical therapies for symptomatic PLD include surgical interventions or several drugs that reduce intracellular cyclic adenosine monophosphate (cAMP). Although surgical interventions [10–12] can reduce the mass effects of PLD, these treatments have little impact on the progression of PLD, whereas somatostatin analogs (SAs) are approved for clinical use and have been proven to decrease the growth rate of PLD [9, 13–18].

Another drug that reduces intracellular cAMP is the vasopressin 2 receptor (V2R) antagonist tolvaptan, which decreases the progression of ADPKD, suppresses the decline in renal function [3, 19, 20], and has the potential to prolong the time before ESKD onset [21]. Although the influence of tolvaptan on PLD has not yet been fully understood, some case reports showed that hepatic cyst volume decreased after taking tolvaptan [22, 23]. Therefore, tolvaptan could be a candidate for additional medical intervention in addition to the current treatment strategy. As far as we know, there have been no observational studies describing the changes in both the total liver volume (TLV) and total kidney volume (TKV) in each patient, or describing the influence of tolvaptan on PLD in ADPKD. Therefore, we conducted a retrospective cohort study to investigate the natural history of changes in TLV and TKV and analyze the effect of tolvaptan on the PLD growth rate in patients with ADPKD, as well as to identify the prognostic factors associated with treatment effectiveness.

## Materials and methods

### Study design

To evaluate the effect of tolvaptan on PLD in patients with ADPKD, we conducted two types of analyses. First, we compared the changes in the annual growth of the total liver volume (TLV) between patients taking tolvaptan (tolvaptan group) and without tolvaptan (non-tolvaptan group). Second, we compared the changes in the annual growth of the total liver volume (TLV) before and after taking tolvaptan in tolvaptan group. To investigate the prognostic factors associated with the suppressive effect of tolvaptan on the growth rate of TLV, the tolvaptan group was classified into responders and non-responders according to the adjusted change in growth rate after tolvaptan use.

The inclusion criteria of the cohort were patients who visited Toranomon hospital and Toranomon hospital Kajigaya between January 2012 and December 2019, were over 20 years old, were diagnosed with ADPKD [24] and PLD [1], met the Japanese criteria of tolvaptan use for ADPKD (TKV > 750 mL, annual growth rate of TKV ≥ 5% per year), and whose TLV was greater than 2,000 mL. The exclusion criteria were patients who had chronic liver dysfunction, had difficulty consuming adequate water, were pregnant, or did not provide written informed consent. In addition, patients who received physical interventions for PLD, such as percutaneous cyst drainage, cyst fenestration, and trans-arterial embolization (TAE) of the hepatic artery were excluded in the primary analysis, while patients who received physical interventions for PLD less than one year before enrollment was included in the secondary analysis.

The standard dose of tolvaptan was 60 mg per day, which was increased to 120 mg if increasing dosing found was to be safe and tolerable, and the duration of tolvaptan use was more than 6 months. The prescription of tolvaptan was implemented as standard-of-care throughout the study period and the assignment of patients to tolvaptan was made at the discretion of the respective treating physician. All data were retrieved from the database at Toranomon hospital and they were fully anonymized before we accessed them. This study followed the principles of the Helsinki declaration and was approved by the Toranomon Hospital Institutional Review Board in 2017 (IRB number 1475). The IRB waived the requirement for informed consent as this was a retrospective study.

## Variables and outcome

The clinical data included sex, various dates (enrollment, diagnosis, computed tomography (CT), start and cessation of tolvaptan), BMI, menopausal status (assessed by the self-reported questionnaire), comorbidities (hypertension, and diabetes mellitus), previous medical history (angiotensin converting enzyme inhibitors (ACEis), angiotensin II receptor blockers (ARBs), or ursodeoxycholic acid (UDCA)), history of physical interventions for PLD, and menstrual status self-reported by patients. Laboratory data included albumin, aminotransferase, alkaline phosphatase, gamma-glutamyl transpeptidase, total-bilirubin, partial thromboplastin time, creatine, and glomerular filtration rate as estimated by the Japanese equation [25]. Laboratory data were obtained within 1 month of when the first liver imaging was conducted. We applied CTs performed during the annual checkups for renewing intractable disease, and applied these data to calculate the TLV and TKV; however, there was no additional exposure to radiation during this study. The CTs were performed with 5 mm slices without contrast media. Volumetry was performed by using the 3D image analysis system, "SYNAPSE VINCENT", which semiautomatically traces the area of the targeted organs of each slice and integrates each slice volume. The tracing was conducted by two technicians (C.N., and Y.M.) who did not know the response to tolvaptan. After double checking by H.M. and Y.U., each TLV was defined as the mean value of the TLVs traced by the two technicians. These data were inspected by the data monitoring and independent audit committee.

The non-tolvaptan period in the tolvaptan group was defined as the time between the start of observation and the last CT before the initiation of tolvaptan, while the tolvaptan period in the same group was defined as the time between the first day of tolvaptan and the time of the last CT before the cessation of tolvaptan or December 2019, whichever came first. Whereas, in the non-tolvaptan group, we divided the observational period into the former and latter observational period in order to compare the outcomes of the tolvaptan group. To minimize the carry-over effect of other interventions, if patients experienced any other intervention for PLD, data for one year after such treatment was excluded from both observational periods. In addition, we set up the dataset that consisted of the patients who had no history of interventions.

The primary outcome was defined as the change from the baseline annual growth rate of TLV ($\Delta$TLV%), which was calculated as the absolute subtraction of growth rate of TLV in the non-tolvaptan period of tolvaptan group from that in tolvaptan period or of growth rate of TLV in the former observational period of the non-tolvaptan group to that in latter observational period. The annual growth rates of TLV and TKV were defined as the slope as estimated by using a linear model from more than two measurements in each period. In secondary analysis, to analyze the prognostic factors, we divided the tolvaptan group into two groups according to their $\Delta$TLV%. Responders were defined as those with negative $\Delta$TLV%, while non-responders were defined as those with a positive $\Delta$TLV%.

## Statistical analysis

The categorical data were described as numbers and percents and were analyzed by chi-square tests or two-tailed Fisher's exact test as appropriate, while continuous variables which were normal distribution were described as the mean and standard deviation and were analyzed by student's t-test and continuous variables that is not normal distribution are described as the median, minimum, and maximum and were analyzed by Wilcoxon rank-sum test. Coefficient of correlation was evaluated by Spearman's rank correlation test. Any p-values below 0.05 were considered significant. To determine the suppressive effect of tolvaptan on TLV, logistic regression models were implemented. The adjusted factors in the logistic regression analysis were age, sex, body mass index (BMI), mean blood pressure, height adjusted TKV (HtTKV), maintenance dose of tolvaptan and variables for which the p-values were below 0.05. All statistical analyses were conducted by using R version 3.4.3 (R Foundation for Statistical Computing, Vienna, Austria).

## Results

### Baseline characteristics

The study flow chart is shown in Fig 1. Between January 2012 and December 2019, tolvaptan was prescribed to 295 out of 902 ADPKD patients with preserved kidney function (eGFR>15 mL/min/1.73 m²) who visited our hospitals due to PLD or ADPKD. Among them, a total of 46 cases were eligible for the tolvaptan group and 16 patients were for the non-tolvaptan group. In the tolvaptan group, twenty-three patients (50.0%) were male and the mean patient age was 51.8 years old. The median height adjusted TLV (HtTLV) was 1068 (range 557–6691) mL/m, while the mean HtTKV was 994 (450–4152) mL/m. Hypertension was diagnosed in 36 out of 46 (78.3%) patients. The proportions of patients using ACEi/ARB, and UDCA were 65.2% and 2.2%, respectively. Laboratory data showed that the mean and SD of eGFR was 50.5±20.2 mL/min/1.73 m². Forty percent of patients had proteinuria with a median of 0.08 (0.01, 1.69) g/

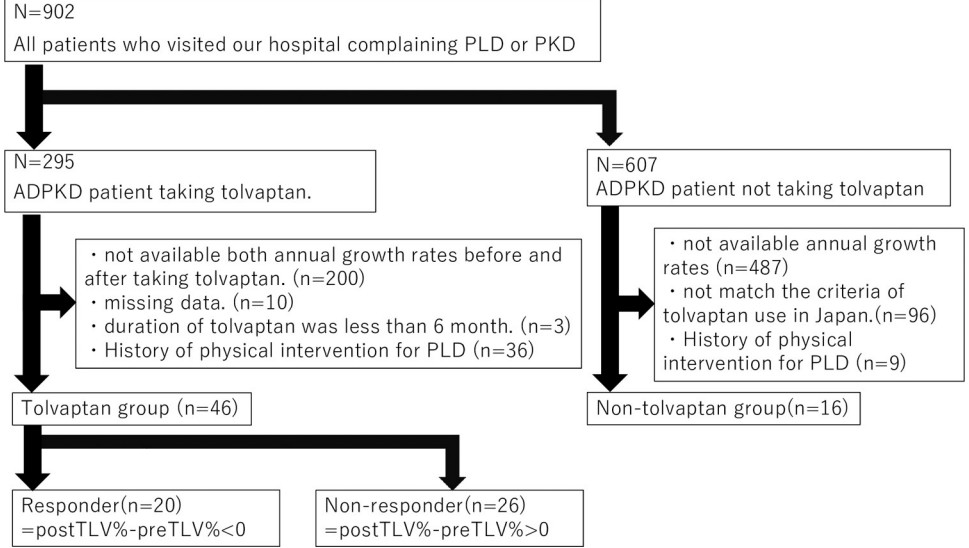

**Fig 1. Flow chart of patients.** In total 902 patients, tolvaptan was prescribed for 295 patients. Among them, 46 patients were eligible for the tolvaptan group. In the other 607 patients, 16 patients were eligible for the non-tolvaptan group. PLD: polycystic liver disease, PKD: polycystic kidney disease, ADPKD: autosomal dominant polycystic kidney disease, TLV%: annual growth rate of total liver volume.

**Table 1. The baseline demographic and laboratory data of the tolvaptan group and non-tolvaptan group.**

| | | Tolvaptan group | | Control | | | |
|---|---|---|---|---|---|---|---|
| | | n = 46 | | n = 16 | | p value | |
| **Baseline characteristics** | | | | | | | |
| Male | n(%) | 23 | (50.0) | 10 | (62.5) | 0.39 | |
| Age | (y.o.) | 51.8 | ±10.3 | 48.7 | ±14.4 | 0.45 | |
| Height | (cm) | 166.2 | ±9.7 | 166.1 | ±12.0 | 0.99 | |
| Body weight | (kg) | 63.5 | ±11.9 | 62.2 | ±10.0 | 0.68 | |
| Body-mass index | (kg/m$^2$) | 22.9 | ±3 | 22.6 | ±2.9 | 0.72 | |
| Systolic blood pressure | (mmHg) | 127.2 | ±15 | 121.8 | ±11.9 | 0.18 | |
| Diastolic blood pressure | (mmHg) | 80.8 | ±10.4 | 77.7 | ±11.5 | 0.39 | |
| Height adjusted total liver volume | (mL/m) | 1068 | (557–6691) | 918 | (640–4177) | 0.38 | |
| Height adjusted total kidney volume | (mL/m) | 994 | (450–4152) | 513 | (405–1928) | **<0.01** | ** |
| **Comorbidities, n(%)** | | | | | | | |
| Hypertension | n(%) | 36 | (78.3) | 12 | (75.0) | 0.79 | |
| Diabetes mellitus | n(%) | 2 | (4.3) | 0 | (0.0) | 0.40 | |
| **Medications, n(%)** | | | | | | | |
| Angiotensin converting enzyme inhibitor or angiotensin II receptor blocker | n(%) | 30 | (65.2) | 10 | (62.5) | 0.84 | |
| Ursodeoxycholic acid | n(%) | 1 | (2.2) | 0 | (0.0) | 0.55 | |
| **Laboratory values (serum)** | | | | | | | |
| Platelet count | ($^*10^3$/μL) | 228.6 | ±61.7 | 243.1 | ±75.3 | 0.60 | |
| Albumin | (g/dL) | 3.9 | ±0.4 | 4.3 | ±0.4 | **<0.01** | ** |
| Aspartate aminotransferase | (IU/L) | 18.8 | ±5.6 | 21.7 | ±5.4 | 0.09 | |
| Alanine aminotransferase | (IU/L) | 16.0 | ±8.4 | 18.3 | ±7.6 | 0.32 | |
| Alkaline phosphatase | (IU/L) | 190.2 | ±74.8 | 174.8 | ±63.4 | 0.44 | |
| Gamma glutamyltransferase | (IU/L) | 35.4 | ±37.2 | 36.6 | ±31.9 | 0.90 | |
| Total bilirubin | (mg/dL) | 0.7 | ±0.2 | 0.7 | ±0.2 | 0.48 | |
| Uric acid | (mg/dL) | 6.2 | ±1.4 | 5.6 | ±1.7 | 0.27 | |
| Creatinine | (mg/dL) | 1.2 | ±0.4 | 0.9 | ±0.2 | **<0.01** | ** |
| eGFR | (mL/min/1.73m$^2$) | 50.5 | ±20.2 | 73.7 | ±23.5 | **<0.01** | ** |
| Prothrombin time | (%) | 103.3 | ±9.6 | 101.9 | ±10.4 | 0.73 | |
| **Laboratory values (urine)** | | | | | | | |
| Hematuria | n(%) | 6 | (13) | 2 | (12.5) | 0.96 | |
| Proteinuria | (g/gCre) | 0.08 | (0.01–1.69) | 0.06 | (0.01–0.25) | 0.15 | |
| N-acetyl-D-glucosamine | (U/mL) | 5.1 | ±2.9 | 4.5 | ±1.8 | 0.44 | |

gCre, and 13.0% had hematuria. Their mean serum albumin was 3.9±0.4 g/dL, their mean total bilirubin was 0.7±0.2 g/dL and all patients were classified as Child-Pugh A. Comparing to the non-tolvaptan group, tolvaptan group had larger mean HtTKV (p<0.01), lower serum albumin level (p<0.01), and higher serum creatinine level (p<0.01) (Table 1).

## Exposure

During the tolvaptan period, the mean duration of tolvaptan use was 3.4±1.34 years. The mean maintenance dose was 65.1±29.7mg per day. The cessation of tolvaptan use was found in 11 out of 46 patients for the following reasons: missing the follow up (n = 2), declining the prescription (n = 1), liver dysfunction (n = 3), polyuria (n = 2), and kidney dysfunction below eGFR 30 mL per minutes per 1.73 meter square (n = 3), which did not affect the growth rate of TLV and TKV.

## Analysis of the whole cohort

In total, 359 CT examinations among 46 patients of tolvaptan group and 71 CT examinations among 16 patients of non-tolvaptan group were used to calculate the TLV. In the tolvaptan group, the mean and SD of the examinations of each patient were 2.8±1.2 in the non-tolvaptan period and 5.0±1.8 in the tolvaptan period. The mean duration between the first CT and the last CT in the non-tolvaptan period was 2.3±1.2 years, and in the tolvaptan period it was 2.2 ±0.9 years. Whereas, in the non-tolvaptan group, the mean and SD of the examinations of each patient were 2.6±1.0 in the former observational period and 2.7±0.6 in the latter observational period. The mean duration of the former observational period in the non-tolvaptan group was 2.6±1.0 years, and that of the latter observational period was 2.4±1.5 years.

**Annual growth rate of TLV and TKV.** In tolvaptan group, the median growth rate of TLV in the non-tolvaptan period and in the tolvaptan period were 1.2 (range -6.9, 16.3) %/year and 2.4 (-8.6, 17.6) %/year, respectively, with a median difference in the annual change in TLV of -0.8 (-15.9, 16.7) %/year (p = 0.78). On the contrary, in the non-tolvaptan group, the median growth rate of TLV in the former observational period and in the latter observational period were 0.2 (-4.8, 15.6) %/year and 1.2 (-5.3, 17.8) %/year, respectively, with median difference in the annual change in TLV of 2.0 (-15.6, 16.6) %/year. The median difference in the annual change in TLV was not statistically different between the tolvaptan group and the non-tolvaptan group (p = 0.52) (Fig 2).

On the other hand, the median difference of the annual change in TKV of tolvaptan group and the non-tolvaptan group were 0.2 (-39.3, 22.5) %/year and 0.7 (-7.6, 23.4) %/year, respectively, which was not statistically different between groups (p = 0.39) (Fig 3).

**Analysis of the prognostic factors of annual growth rate of TLV.** To identify the predictors of response to tolvaptan, secondary analysis was conducted in the tolvaptan group. As shown in Fig 4, 20 out of 46 patients (43.5%) experienced a decline in the ΔTLV% after taking tolvaptan, we defined them as responders. The other 26 patients (56.5%), who experienced an increase in ΔTLV% after taking tolvaptan, were categorized as non-responders. Compared with the non-responders, responders had a higher annual change in TLV in the non-tolvaptan period (4.5 (-3.2–16.3) vs -0.7 (-6.9–5.2) percent per year, p<0.01). Differences in HtTLV, kidney function, liver function and dose of tolvaptan were not statistically significant between the groups (Table 2). The daily dose of tolvaptan (S1 Fig) and menopausal status (S2 Fig) were not significantly different between responder and non-responder and both were not associated with ΔTLV% and responder rate.

**Prognostic factors of annual growth rate of TLV.** In logistic regression analysis, the unadjusted model showed that older age (odds ratio 3.00 [95% CI 1.37–6.55, p<0.01) and a higher growth rate of TLV at baseline (odds ratio 1.41 [95% CI 1.13–1.75], p<0.01) were associated with the responders. After adjusting for baseline variables including sex, age, BMI, mean blood pressure, height adjusted TKV, annual change in TLV in the non-tolvaptan period, a medical history of UDCA and maintenance dose of tolvaptan, older age (odds ratio 1.15 [95% CI 1.01–1.32]) and a higher growth rate of TLV in the non-tolvaptan period (odds 1.45 95% CI 1.10–1.90) were significantly associated with the responders (Table 3).

## Analysis of the patients including the history of physical interventions to the polycystic liver

In order to re-evaluate the prognostic factors of tolvaptan, we conducted the analysis of the cohort including patients with the physical interventions of polycystic liver, such as trans-arterial embolization of liver (liver TAE) and liver cyst drainage. In this cohort, eighty-two patients were eligible to the tolvaptan group and twenty-five patients were eligible to the non-tolvaptan

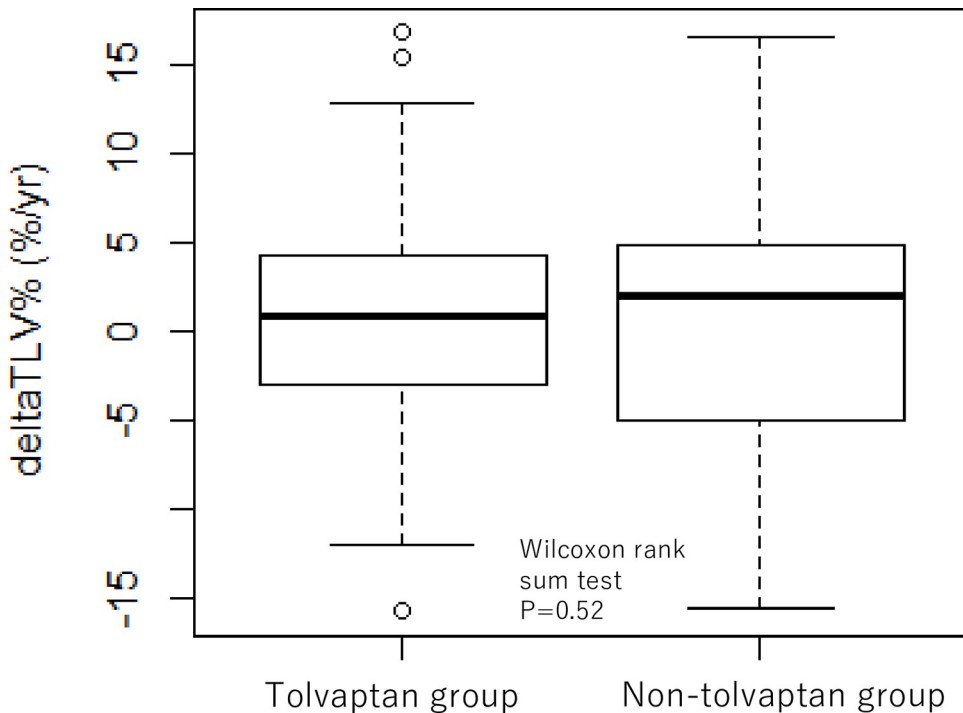

**Fig 2. Comparison of the change in annual growth rate of total liver volume between the tolvaptan group and the non-tolvaptan group.** The median difference in the annual change in TLV was not statistically different between the tolvaptan group (-0.8 (-15.9, 16.7) %/year) and the non-tolvaptan group (1.7 (-15.6, 18.7)) (p = 0.52, Wilcoxon rank sum test). TLV; total liver volume, ΔTLV %; change in annual growth rate of TLV.

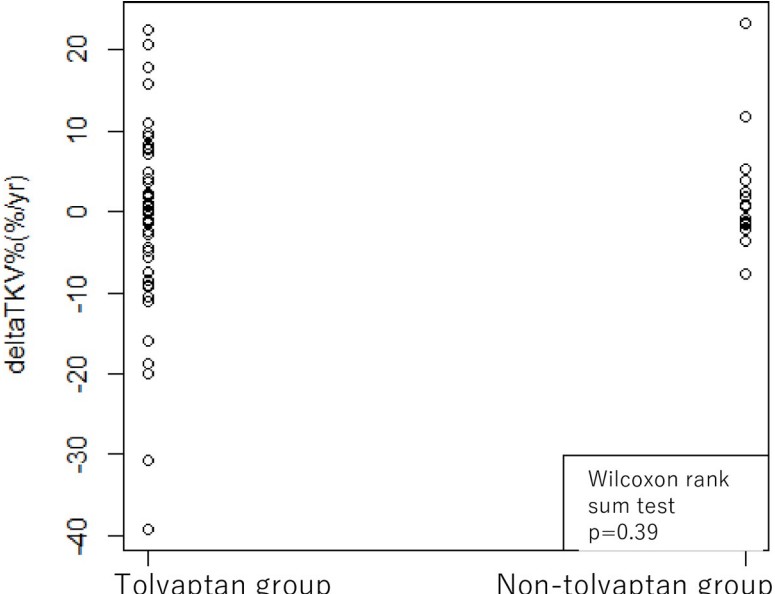

**Fig 3. Comparison of the change in annual growth rate of total kidney volume between the tolvaptan group and the non-tolvaptan group.** The median difference in the annual change in TKV was not statistically different between the tolvaptan group (0.2 (range -30.3, 22.5)) and the non-tolvaptan group (2.0 (-15.6, 16.6)) (p = 0.39, Wilcoxon rank sum test). TKV: total kidney volume, ΔTKV%: the change in growth rate of TKV.

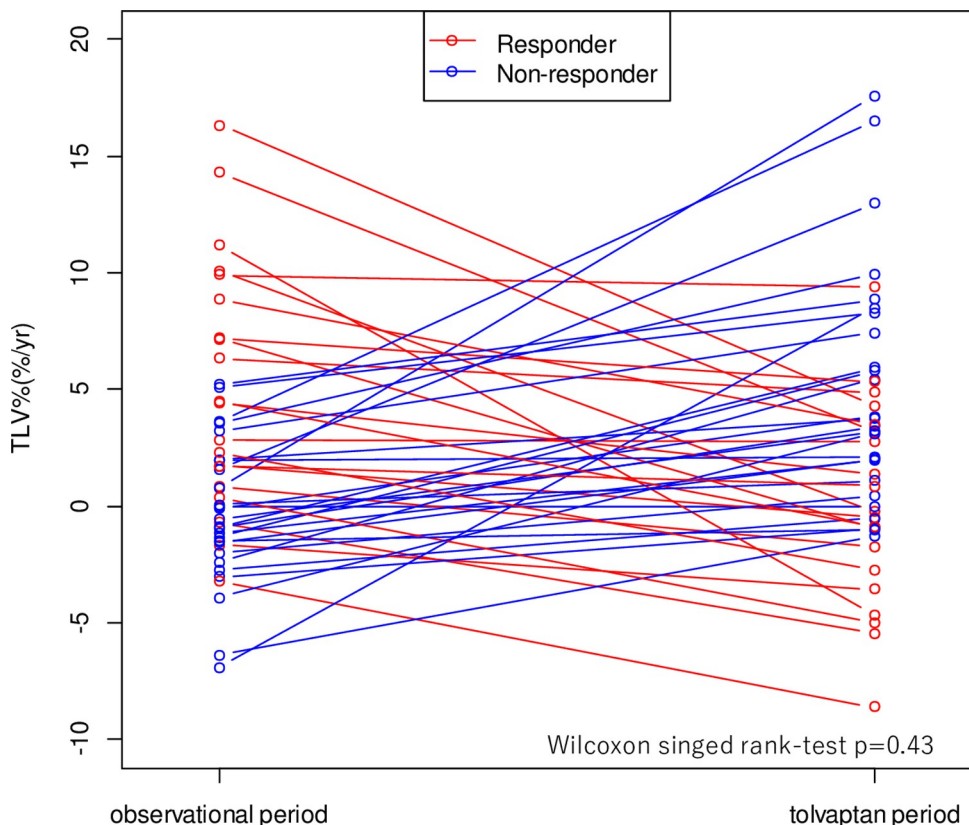

**Fig 4. The spaghetti plotting of the annual liver growth rate before and after tolvaptan use.** In the tolvaptan group, the median growth rate of TLV before tolvaptan use was 1.23 (range -6.9, 16.3) %/year and that after tolvaptan use was 2.4 (-8.6, 17.6) %/year. Although, the growth rate of TLV did not statistically change after tolvaptan use (p = 0.78, Wilcoxon test for paired observations), 20 out of 46 patients (43.5%) experienced a decline in the change in annual growth rate of TLV (ΔTLV%) after taking tolvaptan, we defined them as responders (red circle). The other 26 patients (56.5%), who experienced an increase in ΔTLV% after taking tolvaptan, were categorized as non-responders (blue circle). TLV: total liver volume, ΔTLV%: change in annual liver growth rate of TLV.

group. Even in this setting, patients in the tolvaptan group had large htTKV, higher creatinine level, lower eGFR and lower serum albumin level than in those who in the non-tolvaptan group (S1 Table). In addition, 25 (30.5%) patients had a history of liver TAE and 19 (23.2%) patients had a history of liver cyst drainage. The median volume of cyst drainage was 656 (140–2575) in the tolvaptan group, which was smaller than that in the non-tolvaptan group (1100(450–3385), p-value<0.01). Both interventions were conducted more than one year before the observational period.

**Change in annual growth rate of TLV and TKV.** The median ΔTLV% was not statistically different between the tolvaptan group (0.7 (-35.0, 54.2) %/year) and the non-tolvaptan group (1.7 (-15.6, 18.7) %/year) (p = 0.29) (S3 Fig). Whereas, the median ΔTKV% in tolvaptan group and the non-tolvaptan group were 0.2 (-42.0, 50.9) %/year and 1.1 (-55.0, 23.4) %/year, respectively and the reduction of annual growth rate in TKV was larger in tolvaptan group than in the control group (p = 0.02) (S4 Fig).

**Prognostic factors of annual growth rate of TLV.** In the tolvaptan group of this subgroup, forty-four patients (53.7%) were the responders, who had more rapid progression of annual growth rate of TLV than non-responders at baseline (S2 Table). In logistic regression analysis adjusted by related variables (sex, age, body-mass index, height adjusted total kidney

**Table 2. The baseline demographic and laboratory data of responders and non-responders.**

| | | Responder | | Non-responder | | | |
|---|---|---|---|---|---|---|---|
| | | n = 20 | | n = 26 | | p value | |
| Baseline characteristics | | | | | | | |
| Male | n(%) | 9 | (45.0) | 14 | (53.8) | 0.67 | |
| Age | (years old) | 57.2 | ±9.9 | 47.6 | ±8.6 | 0.09 | |
| Height | (cm) | 163.5 | ±9.0 | 168.2 | ±9.8 | 0.24 | |
| Body weight | (kg) | 62.1 | ±11.1 | 64.6 | ±12.4 | 0.64 | |
| Body-mass index | (kg/m$^2$) | 23.1 | ±2.9 | 22.7 | ±3.0 | 0.58 | |
| Systolic blood pressure | (mmHg) | 129.2 | ±13.2 | 125.7 | ±16.1 | 0.38 | |
| Diastolic blood pressure | (mmHg) | 80.0 | ±9.4 | 81.4 | ±11.0 | 0.81 | |
| Height adjusted total liver volume | (mL/m) | 1170 | (629–6691) | 1007 | (557–3380) | 0.39 | |
| Height adjusted total kidney volume | (mL/m) | 1033 | (477–4152) | 979 | (450–3515) | 0.55 | |
| Annual growth rate of TLV | (%/year) | 4.5 | (-3.2–16.3) | -0.7 | (-6.9–5.2) | <0.01 | ** |
| Annual growth rate of TKV | (%/year) | 7.5 | (-4.7–46.6) | 2.7 | (-12.8–19.8) | 0.17 | |
| post menopausal female | n(%) | 7 | (87.5) | 5 | (83.3) | 0.61 | |
| Dose of tolvaptan | (mg/day) | 57.4 | ±32.5 | 72.1 | ±27.3 | 0.12 | |
| Comorbidities | | | | | | | |
| Hypertension | n(%) | 17 | (85.0) | 19 | (73.1) | 0.25 | |
| Diabetes mellitus | n(%) | 1 | (5.0) | 1 | (3.8) | 0.81 | |
| Medications | | | | | | | |
| Angiotensin converting enzyme inhibitor or angiotensin II receptor blocker | n(%) | 15 | (75.0) | 15 | (57.7) | 0.08 | |
| Ursodeoxycholic acid | n(%) | 1 | (5.0) | 0 | (0.0) | 0.24 | |
| Laboratory values (serum) | | | | | | | |
| Platelet count | (*10$^3$/μL) | 223.5 | ±78.9 | 232.5 | ±43.8 | 0.29 | |
| Albumin | (g/dL) | 3.9 | ±0.3 | 3.8 | ±0.4 | 0.68 | |
| Aspartate aminotrasferase | (IU/L) | 18.8 | ±4.2 | 18.9 | ±6.5 | 0.39 | |
| Alanine aminotransferase | (IU/L) | 14.3 | ±5.4 | 17.2 | ±9.9 | 0.96 | |
| Alkaline phosphatase | (IU/L) | 202.2 | ±82.9 | 181.1 | ±66.5 | 0.41 | |
| Gamma glutamyltransferase | (IU/L) | 31.4 | ±21.6 | 38.5 | ±45.4 | 0.93 | |
| Total bilirubin | (mg/dL) | 0.7 | ±0.2 | 0.7 | ±0.3 | 0.87 | |
| Uric acid | (mg/dL) | 6.2 | ±1.5 | 6.1 | ±1.3 | 0.47 | |
| Creatinine | (mg/dL) | 1.3 | ±0.6 | 1.2 | ±0.3 | 0.21 | |
| eGFR | (mL/min/1.73m$^2$) | 46.8 | ±21.4 | 53.3 | ±18.7 | 0.26 | |
| Prothrombin time | (%) | 103.0 | ±11 | 103.6 | ±8.2 | 0.74 | |
| Laboratory values (urine) | | | | | | | |
| Hematuria | n(%) | 4.0 | (20.0) | 2 | (7.7) | 0.64 | |
| Proteinuria | (g/gCre) | 0.11 | (0.01–1.69) | 0.08 | (0.02–0.55) | 0.75 | |
| N-acetyl-D-glucosamine | (U/mL) | 5.9 | ±3.1 | 4.4 | ±2.6 | 0.27 | |

volume, annual change of total liver volume in the non-tolvaptan period, history of taking ursodeoxycholic acid, volume of drainage), only baseline annual growth rate in total liver volume (odds 1.17 [95% CI 1.01–1.36], p = 0.04) were significantly associated with the responders (S3 Table).

## Discussion

In this study, the change in the annual growth rate of TLV was not statistically significant between the tolvaptan group and the non-tolvaptan group. However, we found that more than half of the tolvaptan group experienced a decrease in the annual growth rate of TLV after the

**Table 3. Logistic regression model analyzing the prognostic factors of the growth rate of TLV in ADPKD.**

| Predictors | | Unadjusted | | | | Adjusted | | | |
|---|---|---|---|---|---|---|---|---|---|
| | | Odds | 95% CI | p-value | | Odds | 95% CI | p-value | |
| Male | | 0.70 | (0.22–2.26) | 0.55 | | 5.55 | (0.45–68.97) | 0.18 | |
| Age | (10years old) | 3.00 | (1.37–6.55) | **0.01** | * | 1.15 | (1.01–1.32) | **0.04** | * |
| Body-mass index | (kg/m$^2$) | 1.05 | (0.86–1.28) | 0.63 | | 0.93 | (0.65–1.34) | 0.69 | |
| Mean blood pressure | (mmHg) | 1.00 | (0.95–1.06) | 0.95 | | 0.98 | (0.91–1.06) | 0.68 | |
| Height adjusted total kidney volume | (100mL/m) | 1.03 | (0.96–1.11) | 0.41 | | 0.94 | (0.82–1.07) | 0.33 | |
| Annual change of total liver volume | (%/year) | 1.41 | (1.13–1.75) | **<0.01** | ** | 1.45 | (1.1–1.9) | **0.01** | * |
| Ursodeoxychcolic acid | | $7.88{*}10^6$ | $(0.00-e^{2868.45})$ | 0.99 | | $2.65{*}10^7$ | $(0.00-e^{4720.20})$ | 0.99 | |
| Dose of tolvaptan | (30mg/day) | 0.98 | (0.96–1.00) | 0.11 | | 0.98 | (0.96–1.01) | 0.29 | |

Abbreviations. ADPKD: autosomal dominant polycystic kidney disease. TLV: total liver volume.

initiation of tolvaptan use and these patients had a more rapid TLV growth rate at baseline. After adjusting for related variables, older age and a rapid progression of TLV were associated with the responders to tolvaptan.

V2R is now known to be more widely expressed in the kidney, including thick ascending limb of Henle [26, 27]. In the polycystic kidney, the concentration of intracellular cAMP is increased and plays a key role in the progression of ADPKD [28]. In the polycystic liver, cAMP is also a main target signal of SAs, which reduces the intracellular cAMP via the somatostatin receptor, which may decrease proliferation in the polycystic liver [9, 13–18]. Additionally, a recent study showed that V2R was expressed on cholangiocytes from patients with ADPKD and the proliferation of these cholangiocytes was inhibited by a V2R antagonist in vitro [29]. These findings suggest that tolvaptan would be expected to slow the progression of PLD in the clinical setting. In this observational study, we noticed that there were a certain number of responders whose TLV growth rates were decreased after tolvaptan use, although the change in annual growth rate of TLV was not different between the groups and the median growth rate of TLV seemed almost consistent before and after tolvaptan use overall.

We observed that older age was a factor associated with responders to tolvaptan in this study, while female sex was not. Previous studies showed that younger age [8, 9], larger HtTLV [8], and female sex [9] had a higher annual growth rate of liver. In this study, we chose the inclusion criteria including Japanese criteria of tolvaptan use for ADPKD and younger female patients aged under 45 years old was only 8.7 percent and premenopausal female patients was only 14.3 percent of all female patients in the tolvaptan group. Therefore, less active polycystic liver disease might have been included in our study. These baseline differences could reduce the response to tolvaptan.

Another predictor of decreasing PLD growth rate was a higher baseline PLD growth rate. In a previous randomized control study, patients with a higher liver growth rate responded well to SAs [9]. In addition, in patients with ADPKD, a higher percentage with progression to TKV showed the greater effects of tolvaptan on TKV [3, 30, 31], which suggested that a higher growth rate would be a good prognostic marker for medical interventions related to cAMP. Although the change in annual liver growth rate was not statistically different, the secondary analysis for the tolvaptan group were consistent with those of previous studies, however, careful interpretations will be necessary to apply these results in clinical practice.

The spontaneous reduction of polycystic liver was previously reported by [8, 9] and 30–44% of the patients without any intervention. Majority of these patients were over 48 years old [8]. In this study, mean age in our patients was older than previous studies, and the reduction

of annual growth rate of TLV could be partially explained by this spontaneous reduction. However, logistic analysis statistically showed that the higher annual growth rate of TLV was the predictor of reduction of annual liver growth rate, meaning the disease-modifying effect of tolvaptan other than the spontaneous reduction was suggested.

The strength of this study is that this is the first observational study to evaluate the effect of tolvaptan on PLD using a relatively large number of patients with PLD as compared to previous studies. Furthermore, to minimize the intraobserver and interobserver variance, we used multiple CT examinations with a semiautomated system to calculate the estimated liver growth. Although our cohort only consisted of liver-cyst dominant patients with ADPKD, such characteristics were commonly found in previous studies of PLD due to ADPKD [9, 13–18, 29], and medical therapy other than SAs is an unmet need for these patients.

Several limitations still exist in our study. First, this is the retrospective observational study and cause-and-effect of the results was not determined. Second, because of the inclusion criteria of this study, external validity of our results was limited if the use of tolvaptan was considered to the patients of younger female patients with large PLD without ADPKD or liver dominant PLD with ADPKD. Third, the genetic evaluation was not conducted, which meant that some genetic heterogeneity might have existed in our cohort. However, it has been reported that the genotype of ADPKD does not influence the severity or progression of PLD [8]. Fourth, the change of annual growth rate of TKV was not significantly different between tolvaptan group and non-tolvaptan group. This might suggest that the less active patients were included in terms of polycystic kidney disease. Therefore, more active ADPKD patients having both kidney and liver should have been selected. Finally, although sex did not predict the suppressive effects of tolvaptan on the PLD growth, the expression of estrogen and its receptor on cholangiocytes have been shown to be related to the growth rate of the liver [32]. Furthermore, in our study, information about the number of births and the duration of exposure to oral contraceptives were not collected from the medical chart. Although menopausal status was collected from only about half of female patients in the tolvaptan group, subgroup analysis about menopausal status showed it did not change the response of tolvaptan. Hence, additional well-designed prospective studies are necessary to analyze the effect of tolvaptan on PLD.

In conclusion, compared to the non-tolvaptan group, the annual growth rate in total TLV in ADPKD patients taking tolvaptan did not decrease in the entire cohort. However, subgroup analysis revealed that 43.5% in the tolvaptan group experienced the reduction of the annual growth rate of TLV and after adjusting for several related factors, older age and a higher growth rate of TLV were associated with the reduction of the annual growth rate of TLV. PLD is one of the intractable diseases with limited treatment options, and the mass effects of PLD often reduce patient quality of life and can sometimes be life-threatening. Although some interventional studies are required to confirm the effectiveness of tolvaptan for PLD, our results may help guide clinical practice for PLD patients.

## Supporting information

**S1 Fig. Analysis of dose effect of tolvaptan on polycystic liver disease.** To assess the dose effect of tolvaptan on the change in annual growth rate of total liver volume (TLV), the original tolvaptan group were stratified by daily dose of tolvaptan, less than 30mg, 60mg, 90mg, and mg. However, we could not find the any trends of change in annual growth rate of total liver volume nor response rate associated to the dose of tolvaptan. Abbreviation.ΔTLV% change in annual growth rate of TLV.
(PDF)

**S2 Fig. Spaghetti plotting of annual growth rate of total liver volume before and after tolvaptan use in premenopausal and postmenopausal patients.** The annual growth rate of total liver volume was not change in premenopausal female(n = 2) and in postmenopausal female (n = 12). Red circle represents the responder defined as the patients whose annual liver growth rate of total liver volume decreased after tolvaptan use, while the blue circle represents the non-responder defined as the patients whose annual liver growth rate of total liver volume increased. Abbreviation. TLV%: annual growth rate of total liver volume.
(PDF)

**S3 Fig. Comparison of the change in annual growth rate of total liver volume between the tolvaptan group and the non-tolvaptan group in patients including the history of physical interventions for polycystic liver.** The median change in the annual growth rate of total liver volume (TLV) was not statistically different between the tolvaptan group (-0.7 (range -35.0, 54.2) %/year) and the non-tolvaptan group (1.7 (-15.6, 18.7) %/year) (p = 0.29, Wilcoxon rank sum test). ΔTLV %: change in annual growth rate of TLV.
(PDF)

**S4 Fig. Comparison of the change in annual growth rate of total kidney volume between the tolvaptan group and the non-tolvaptan group in patients including the history of physical interventions for polycystic liver.** The median difference in the annual growth rate in total kidney volume (TKV) was larger in the tolvaptan group (0.2 (-42.0, 50.9)) than in the non-tolvaptan group (1.1 (-55.0, 23.4)) (p = 0.02, Wilcoxon rank sum test). Abbreviation. ΔTKV%: the change in growth rate of TKV.
(PDF)

**S1 Table. The baseline demographic and laboratory data of the tolvaptan group and non-tolvaptan group in patients without the history of interventions for polycystic liver.**
(DOCX)

**S2 Table. The baseline demographic and laboratory data of responders and non-responders in patients without the history of interventions for polycystic live.**
(DOCX)

**S3 Table. Logistic regression model analyzing the prognostic factors of the growth rate of TLV in ADPKD in patients without the history of interventions for polycystic liver.**
(DOCX)

**S1 Dataset.**
(DOCX)

## Acknowledgments

We thank Ms. Yurina Takaishi, the data manager of this study, and Ms. Chiho Nakahara and Ms. Yukiko Momoi for the TLV measurements, and other staff members at Toranomon Hospital Kajigaya for data collection and management. We also thank Mr. Yoshihiro Ishihara and Mr. Tsubasa Suzuki, Flexible Inc., Japan, for data monitoring and forming the independent audit committee. H.M. and J.H. designed and conducted the study, provided the analyses, and wrote the manuscript. A.G. supervised the data analyses. Other coauthors managed the patients and contributed to the discussion.

## Author Contributions

**Conceptualization:** Hiroki Mizuno, Junichi Hoshino.

**Data curation:** Hiroki Mizuno.

**Formal analysis:** Hiroki Mizuno, Junichi Hoshino.

**Funding acquisition:** Hiroki Mizuno, Junichi Hoshino.

**Investigation:** Hiroki Mizuno, Junichi Hoshino.

**Methodology:** Hiroki Mizuno, Junichi Hoshino.

**Project administration:** Hiroki Mizuno, Junichi Hoshino.

**Resources:** Hiroki Mizuno.

**Software:** Hiroki Mizuno.

**Supervision:** Hiroki Mizuno.

**Validation:** Hiroki Mizuno.

**Visualization:** Hiroki Mizuno.

**Writing – original draft:** Hiroki Mizuno.

**Writing – review & editing:** Hiroki Mizuno, Akinari Sekine, Tatsuya Suwabe, Daisuke Ikuma, Masayuki Yamanouchi, Eiko Hasegawa, Naoki Sawa, Yoshifumi Ubara, Junichi Hoshino.

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
