## [Decision Letter · Decision Letter 0]

10 Jun 2021

PONE-D-21-12669

Potential effect of tolvaptan on polycystic liver disease with ADPKD

PLOS ONE

Dear Dr. Mizuno,

Thank you for submitting your manuscript to PLOS ONE. After careful consideration, we feel that it has merit but does not fully meet PLOS ONE’s publication criteria as it currently stands. Therefore, we invite you to submit a revised version of the manuscript that addresses the points raised during the review process.

Sorry for the delayed response.  The experts raised several serious concerns.  Authors should seek for the advice from statistician if the method are proper and type I and II errors are avoided.  Authors should add the statistician on the authorship to share the responsibility.

We look forward to receiving your revised manuscript.

Kind regards,

Tatsuo Shimosawa, M.D., Ph.D.

Academic Editor

PLOS ONE

Journal Requirements:

2. In your ethics statement in the manuscript and in the online submission form, please ensure that you have discussed whether all data/samples were fully anonymized before you accessed them and/or whether the IRB or ethics committee waived the requirement for informed consent. If patients provided informed written consent to have data/samples from their medical records used in research, please include this information.

3. Please include the following information in your Methods:

 1) Whether the use of tolvaptan was implemented as standard-of-care at your hospital throughout the study period.

2) Whether assignment of patients to tolvaptan was made at the discretion of the treating physician or for the purposes of research.

4. Please note that PLOS does not permit references to 'data not shown.' Authors should provide the relevant data within the manuscript, the Supporting Information files, or in a public repository. If the data are not a core part of the research study being presented, we ask that authors remove any references to these data.

5.Thank you for stating the following in the Financial Disclosure section:

"This work was supported by J.H’s competitive research grant from Otsuka Pharm, Japan, and a research grant from the Okinaka Memorial Institute for Medical Research, and H.M’s research grant from Toranomon Hospital. The funders had no role in study design, data collection and analysis, decision to publish, or preparation of the manuscript."

We note that you received funding from a commercial source: Otsuka Pharm

Reviewers' comments:

Reviewer's Responses to Questions

**Comments to the Author**

1. Is the manuscript technically sound, and do the data support the conclusions?

Reviewer #1: Yes

Reviewer #2: Yes

Reviewer #3: Yes

Reviewer #4: Partly

2. Has the statistical analysis been performed appropriately and rigorously? 

Reviewer #1: Yes

Reviewer #2: Yes

Reviewer #3: No

Reviewer #4: No

3. Have the authors made all data underlying the findings in their manuscript fully available?

Reviewer #1: No

Reviewer #2: Yes

Reviewer #3: Yes

Reviewer #4: No

4. Is the manuscript presented in an intelligible fashion and written in standard English?

Reviewer #1: Yes

Reviewer #2: Yes

Reviewer #3: Yes

Reviewer #4: Yes

5. Review Comments to the Author

Reviewer #1: The analysis is simple but appropriate for the conducted analysis.

However, the figures 2-4 look strange. I do not know what information they should provide nor is it explained in the paper. The figures are not related to the numerical analysis but provide a visual overview, which is good to provide. They are just unclear! Please discuss them in detail.

Reviewer #2: PLD is one of the most frequent complications of ADPKD, and is an important intractable disorder with limited treatment option. Previous studies suggested that somatostatin analogue may retard the progression of PLD. Although tolvaptan, a vasopressin V2R antagonist, has been approved to prevent PKD progression, its effect on PLD remains unclear. Till date, several case studies have reported that tolvaptan may be effective in controlling PLD; nonetheless, there are currently no observational studies involving a larger number of patients.

In this study by Mizuno et al., the authors have retrospectively analyzed the effect of tolvaptan on PLD in 82 patients. They examined a total of 667 CT scans, and both total liver volume (TLV) and total kidney volume (TKV) were calculated in an unbiased approach using Synapse Vincent. Although the growth rate of TLV was not altered before and after tolvaptan treatment, the authors found that approximately half of patients showed the reduction in TLV (responders). They went on to demonstrate that several factors such as older age, higher annual change in TLV, and use of UDCA were associated with the responders. A logistic regression analysis after adjustment further demonstrated that older age and a higher growth rate of TLV in non-tolvaptan period were significantly associated with the response to tolvaptan.

Overall, the manuscript is clearly written and methods contain sufficient detail. The study deals with clinically significant problem, with the major strength lying in the fact that the authors compared the TLV before and after tolvaptan treatment in more than 80 patients. Given that there is currently scarce evidence in the literature regarding the effect of tolvaptan on PLD, the manuscript will add an important information that is helpful to clinicians in this field. There are, however, several concerns that need to be addressed.

1. Although there were 902 patients with PLD or PKD in total, less than 10% were analyzed in this study, which may constitute a limitation. Given that there were more than 600 patients who did not take tolvaptan, authors might want to calculate the growth rate of TLV in those patients with CT scans available, and compare the rate with the 82 eligible patients. This analysis would make the study more comprehensive, even if the growth rate is not significantly altered.

2. The authors showed that the median growth rate of TLV did not change after taking tolvaptan. However, it is unclear whether the study participants did show response to tolvaptan in terms of growth rate of TKV.

3. The maintenance dose of tolvaptan in each group was not described in Table 1. Was the dosage not associated with the response in PLD?

4. Figures and figure legends need improvement.

- In Figure 2, there is no explanation what blue and red lines mean. This figure also needs the label (A or B) according to the text.

- Order of Figure 3 and Figure 4 could be the other way around. In the main text, Figure 3 is supposed to show the association between TLV and TKV, whereas Figure 4 should show the delta TLV% before and after tolvaptan.

- Figure 4 (association between PLD and PKD): please add correlation coefficient and p value.

- Legends in each figure need more information so that the readers can understand without referring to the main text.

5. Page 8, line 3. aldosterone > angiotensin II

6. Page 9, line 16 (statistical analysis). Is the statistical analysis two-tailed or one-tailed? Please describe.

7. Page 17, line 15-16; consider revising the sentence. V2R is now known to be more widely expressed in the kidney, including thick ascending limb of Henle (Ref: PMID 32035616 and 17626156).

8. “Height adjust total liver volume” in figure legends could be “Height adjusted total liver volume”.

Reviewer #3: Dear authors,

While the efficacy of Tolvaptan in reducing renal cyst growth has been established, its efficacy in complicating extrarenal cysts, particularly in more frequent hepatic cysts, is unknown at present. This study tried to analyze this point, and it is considered to be a informative study with significant clinical advantages, even if the result is positive or negative. However, there are several issues to be resolved as shown below.

Major

Comment #1

This study was a retrospective observational study design without a control group, and the analysis is restricted to individuals with ADPKD who meet the Japanese National criteria for tolvaptan use. Thus, the effect of tolvaptan on hepatic cysts in patients with less severe or less active PKDs or hepatic cysts in patients with predominant hepatic than renal cysts is not included in the analysis. It should be specified that this study is an analysis restricted to hepatic cysts in subjects with ADPKD who met the criteria for tolvaptan use, including title. Although the significance of this study is clearly understandable, I believe that it is inappropriate to refer directly to the effects of tolvaptan in restrospective single arm observation, not in randomized controlled trial, and that careful representation or description is desirable.

Comment #2

Cases with or without hepatic cyst drainage or TAE are mixed in the study population.

In such cases with physical interventions, organic changes or damages may have been occurred in the pericystic tissues associated with invasions, and it is considered inappropriate to analyse populations that may differ qualitatively. Alternatively, it should be discussed that physical interventions are independent of the effects of cyst growth or tolvaptan. I would like to ask for the opinions of the authors. Even if the sample size is reduced, it is considered that the population should be restricted to patients without those physical interventions, or that the subgroup analyses should be added at the least.

Comment #3

It is of great interest that responder, non-responder analysis showed that the more active PLDs, the more pronounced the cyst reduction after tolvaptan use, as well as their efficacy in renal cysts. The results strongly suggest that a similar mechanism to inhibition or reduction of renal cyst growth by tolvaptan may apply to hepatic cysts.

On the other hand, it is not well understandable that there was no correlation between kidney volume reduction and liver volume reduction in the responder group. The influence of the drainage cases and TAE cases may be concerned, but how do the authors consider this point?

Comment #4

There is a lack of discussion of the impact of spontaneous shrinkage of hepatic cysts.

In Ref-8, as a consequence of a large observational study of PLDs, it is reported that spontaneous reduction of hepatic volume is observed in 30% of cases in the population. Ref-9 has also reported a reduction in hepatic volume in half of placebo group. In this study, the liver volume is reduced in about half of the subjects, but how to consider the involvement of spontaneous reduction should be discussed.

Minor

Comment #1

In a stratified analysis of responder and non-responder, adjustments for variables have been described, however, specific method for this statistical procedure has not been described in the text.

Comment #2

Inconsistencies are seen in figure description in the text and figure numbering (e.g. Fig. 2B in P14L7). In addition, both Fig. 2 and Fig. 4 are not represented in A and B.

Reviewer #4: Polycystic liver disease leads to pain and suffering in many individuals with autosomal dominant polycystic kidney disease. This interesting study examines changes in total liver volume in a cohort of individuals who took tolvaptan and followed longitudinally. About 50% experienced a reduction in liver volume and about 50% experienced a continued increase in liver volume. The paper is well written. However I am concerned about the statistics.

There are number of extreme outlier cases who could have influenced the data analysis. Overall we may be seeing a regression to the mean.

Please mark individuals that had TAE or liver volume reducing procedures on the graph.

Please add figure showing both females that were pre and post menopausal and describe what happened to their total liver volumes.

If there is a positive effect of tolvaptan on total liver volume then one might expect a dose -response effect. No dose information has been provided.

6. PLOS authors have the option to publish the peer review history of their article (what does this mean?). If published, this will include your full peer review and any attached files.

Reviewer #1: No

Reviewer #2: No

Reviewer #3: No

Reviewer #4: No

---

## [Author Response · Author response to Decision Letter 0]

31 Aug 2021

Dear Editor and Reviewers

Thank you for reviewing our manuscript “Potential effect of tolvaptan on polycystic liver disease with ADPKD [PONE-D-21-12669]” and offering valuable advices during the tough situation by COVID19 pandemic. We also appreciate the time and effort you and each of the reviews have dedicated to providing insightful feed back on our paper. 

In addition, thanks to your generous kindness of extending the deadline of resubmission, we have finished answering all the comments. The revised manuscript would be improved over the initial manuscript. 

We have addressed your comments with point-by-point responses and revised the manuscript accordingly. We would like to send two versions, clean and marked one and as well as the ppt file of graphic abstract.

Response to Editor and Reviewers’ comments

Editor:

RESPONSE: We had changed all the file naming according to PLOS ONE’s style requirements. 

2. In your ethics statement in the manuscript and in the online submission form, please ensure that you have discussed whether all data/samples were fully anonymized before you accessed them and/or whether the IRB or ethics committee waived the requirement for informed consent. If patients provided informed written consent to have data/samples from their medical records used in research, please include this information.

RESPONSE: We had added the expression describing fully anonymized before we accessed data on page 7 line 17 and the sentence on page 8 line 1-2. 

3. Please include the following information in your Methods:

1) Whether the use of tolvaptan was implemented as standard-of-care at your hospital throughout the study period

RESPONSE: In this retrospective observational study, tolvaptan was used standard-of-care. We have added the following sentence “The use of tolvaptan was implemented as standard-of care throughout the study period” on Page 7 Line 14-15.

2) Whether assignment of patients to tolvaptan was made at the discretion of the treating physician or for the purposes of research.

RESPONSE: In this retrospective observational study, tolvaptan was prescribed by the treating physician not by the purpose of research. We have added the following sentence “the assignment of patients to tolvaptan was made at the discretion of the treating physician”, on Page 7 Line 16.

4. Please note that PLOS does not permit references to 'data not shown.' Authors should provide the relevant data within the manuscript, the Supporting Information files, or in a public repository. If the data are not a core part of the research study being presented, we ask that authors remove any references to these data.

RESPONSE: After conducting the revising analysis, we considered that this information is not a core part, therefore, we decline the expression ‘data not shown’, on Page 25 Line 13. 

5. Financial Disclosure section:

"This work was supported by J.H’s competitive research grant from Otsuka Pharm, Japan, and a research grant from the Okinaka Memorial Institute for Medical Research, and H.M’s research grant from Toranomon Hospital. The funders had no role in study design, data collection and analysis, decision to publish, or preparation of the manuscript."

We note that you received funding from a commercial source: Otsuka Pharm

RESPONSE: We have amended Competing Interests Statement on cover letter and we have added the sentence you offered. 

Reviewer #1: 

The analysis is simple but appropriate for the conducted analysis.

However, the figures 2-4 look strange. I do not know what information they should provide nor is it explained in the paper. The figures are not related to the numerical analysis but provide a visual overview, which is good to provide. They are just unclear! Please discuss them in detail.

RESPONSE: 

Thank you for pointing out figure 2-4’s roles played on this article. After re consideration during revising process, data of figure 2 and figure 4 did not directly related to the major conclusion on this article. Therefore, we declined resubmitting figure 2 and figure 4. Whereas, since figure 3 was the primary outcome of this study, we decided to keep using it, then the figure number was changed from figure 3 to figure 4 in the revised manuscript. In addition, we added the figure caption on Page 17 Line 12. 

Reviewer #2: 

Thank you for providing us with the suggestive comment on our article. We totally agreed with your comment 1, the necessity of the control group. Although it took an enormous time to find the appropriate control group after screening 902 patients, we found 25 patients who were eligible to the control group. Compared with this control group, we could not find the statistical difference of the change in the annual growth rate of TLV between groups. However, we hope this revising process would strengthen our article and give more meaningful suggestions to the future study. 

In order to answer the all the valuable comments from editors and reviews, we decided to decline figure 2 and 4 of the initial manuscript, then we added 2 figures (figure 2 and figure3) and supporting information including 4 figures and 3 tables. Figure 3 of the initial manuscript left as figure 4.

1. Although there were 902 patients with PLD or PKD in total, less than 10% were analyzed in this study, which may constitute a limitation. Given that there were more than 600 patients who did not take tolvaptan, authors might want to calculate the growth rate of TLV in those patients with CT scans available, and compare the rate with the 82 eligible patients. This analysis would make the study more comprehensive, even if the growth rate is not significantly altered.

RESPONSE: We agreed with the statistical limitation of this original manuscript of single arm observational study. Therefore, we decided to find the control group that would be compared to the tolvaptan group. After screening 902 patients to find the appropriate control group, defined ADPKD patients with polycystic liver and fulfilled Japanese criteria of tolvaptan use, and had more than three times CT scanning. We finally found 25 patients that met the prescription criteria of tolvaptan in Japan and set it as a control group. Compared to this control group, the tolvaptan group were larger height adjusted TKV, higher serum creatinine level, and lower eGFR. We divided the observational period of the control group and compared the difference of annual liver growth rate, but there was no statistical difference. 

We added the method of this analysis on Page 6, Line 15-17, and the result of this analysis from page 15 line 6 to page 16 line 7 including figure 2 of revised manuscript. 

We hope these results will make this article have more statistical robustness. 

2. The authors showed that the median growth rate of TLV did not change after taking tolvaptan. However, it is unclear whether the study participants did show response to tolvaptan in terms of growth rate of TKV.

RESPONSE: Thank you for giving us the important advice to evaluate the effect of tolvaptan on polycystic kidney. The annual change in TKV of the tolvaptan group was 0.2 (-42.0, 50.9) %/year while that of the control group was and -1.1 (-55.0, 23.4) %/year. The effect of tolvaptan on the growth rate of TKV was neither significant in this study. We thought that this fact would be partially derived from the liver-dominant ADPKD and less progressive ADPKD patients were selected in this study. We added this result from page 15 line 16 to page 16 line 1 and on figure 3 as well as this limitation on page 25 line 17. 

3. The maintenance dose of tolvaptan in each group was not described in Table 1. Was the dosage not associated with the response in PLD?

RESPONSE: We added the dose of tolvaptan in Table1 and Table2. We created the four categories according to the dose of tolvaptan, however, there was no trend that suggested the dose effect of tolvaptan on response rate or decline of liver growth rate. We added the result of this subgroup analysis on page 17 line 8 and on Supporting Information Figure S1. 

4. Figures and figure legends need improvement.

- In Figure 2, there is no explanation what blue and red lines mean. This figure also needs the label (A or B) according to the text.

RESPOND: Thank you for pointing out the insufficient contents of the legend. As mentioned above. During the revising process, however, we reconsider that the result of this figure was not directly related to the major conclusion of this study, therefore, we decided to decline this figure.

- Order of Figure 3 and Figure 4 could be the other way around. In the main text, Figure 3 is supposed to show the association between TLV and TKV, whereas Figure 4 should show the delta TLV% before and after tolvaptan.

- Figure 4 (association between PLD and PKD): please add correlation coefficient and p value.

- Legends in each figure need more information so that the readers can understand without referring to the main text.

RESPONSE: 

Thank you for pointing out the insufficient contents of the legend. We added the figure legends on figure 4 (delta TLV% before and after tolvaptan) and added the statistical information into the figure. The result of figure3 (the association between TLV and TKV) was not directly related to the major conclusion of this study, so we decided to decline this figure, 

5. Page 8, line 3. aldosterone > angiotensin II

RESPONSE: We have changed the word above (Page8, line 7). 

6. Page 9, line 16 (statistical analysis). Is the statistical analysis two-tailed or one-tailed? Please describe.

RESPONSE: We conducted two-tailed Fisher’s exact test. We added the expression “two-tailed” in Method section on page 10 line 9.

7. Page 17, line 15-16; consider revising the sentence. V2R is now known to be more widely expressed in the kidney, including thick ascending limb of Henle (Ref: PMID 32035616 and 17626156).

RESPONSE: Thank you for showing us valuable reference articles. We adopted the sentence above from Page22, Line 18 to Page 23, Line 1 and added the reference articles. 

8. “Height adjust total liver volume” in figure legends could be “Height adjusted total liver volume”.

RESPONSE: We have changed the expression in figure legends. 

Reviewer #3: Dear authors,

While the efficacy of Tolvaptan in reducing renal cyst growth has been established, its efficacy in complicating extrarenal cysts, particularly in more frequent hepatic cysts, is unknown at present. This study tried to analyze this point, and it is considered to be a informative study with significant clinical advantages, even if the result is positive or negative. However, there are several issues to be resolved as shown below.

RESPONSE: Thank you for providing us the suggestive comment on our article. 

After reconsidering the design of this study, subgroup analysis for the patients without the history of physical intervention were conducted to minimized their effect on PLD. We hope this revising data will strengthen our article and give more meaningful suggestion to the future study. 

Major

Comment #1

This study was a retrospective observational study design without a control group, and the analysis is restricted to individuals with ADPKD who meet the Japanese National criteria for tolvaptan use. Thus, the effect of tolvaptan on hepatic cysts in patients with less severe or less active PKDs or hepatic cysts in patients with predominant hepatic than renal cysts is not included in the analysis. It should be specified that this study is an analysis restricted to hepatic cysts in subjects with ADPKD who met the criteria for tolvaptan use, including title. Although the significance of this study is clearly understandable, I believe that it is inappropriate to refer directly to the effects of tolvaptan in restrospective single arm observation, not in randomized controlled trial, and that careful representation or description is desirable.

RESPONSE: We agreed with your suggestion to change more suitable title for this study. In this revised manuscript, we conducted the analysis by using control group, however, several limitation still had existed and the patients included in this study was limited within the ADPKD patients who met the Japanese criteria of tolvaptan use. Therefore, we changed the title to “Potential effect of tolvaptan on polycystic liver disease for patients with ADPKD meeting the Japanese criteria of tolvaptan use”. 

We hope this title would be more precisely described and more suitable to be accepted.

Comment #2

Cases with or without hepatic cyst drainage or TAE are mixed in the study population.

In such cases with physical interventions, organic changes or damages may have been occurred in the pericystic tissues associated with invasions, and it is considered inappropriate to analyse populations that may differ qualitatively. Alternatively, it should be discussed that physical interventions are independent of the effects of cyst growth or tolvaptan. I would like to ask for the opinions of the authors. Even if the sample size is reduced, it is considered that the population should be restricted to patients without those physical interventions, or that the subgroup analyses should be added at the least.

RESPONSE: We agree with subgroup analysis for the patients without the history of the physical intervention. After excluding the physical intervention, such as liver trans-arterial embolization and drainage for liver cyst, 46 patients were eligible for analysis. In this group, there was no significant difference of annual liver growth rate before and after the tolvaptan use, however, 20(43.4%) patients had experienced the reduction of liver growth rate, responder. Multivariable regression analysis adjusted related variables showed that age and baseline liver growth significantly predicted the responder. We deleted the carry-over effect of trans-arterial embolization of liver from the paragraph about limitation of this study in the discussion part, then we added these results as subgroup analysis from page 21, line 2 to page 22, line 8 and on Supporting Information. 

We believe that these subgroup analysis will more efficiently exclude the carry-over effect of physical intervention and will make our result more clearly understandable. 

.

Comment #3

It is of great interest that responder, non-responder analysis showed that the more active PLDs, the more pronounced the cyst reduction after tolvaptan use, as well as their efficacy in renal cysts. The results strongly suggest that a similar mechanism to inhibition or reduction of renal cyst growth by tolvaptan may apply to hepatic cysts.

On the other hand, it is not well understandable that there was no correlation between kidney volume reduction and liver volume reduction in the responder group. The influence of the drainage cases and TAE cases may be concerned, but how do the authors consider this point?

RESPONSE: 

As we mentioned in the response to comment#2, we conducted subgroup analysis for the patients without the history of the physical intervention. Although the rate of the responder was reduced after excluding these interventions, the prognostic factors related to the responders were the same as that of the main analysis. The result of the main analysis contained such carry-over effects related to the physical intervention, however, we believe this well-designed subgroup analysis minimized this concerned effect. 

Comment #4

There is a lack of discussion of the impact of spontaneous shrinkage of hepatic cysts.

In Ref-8, as a consequence of a large observational study of PLDs, it is reported that spontaneous reduction of hepatic volume is observed in 30% of cases in the population. Ref-9 has also reported a reduction in hepatic volume in half of placebo group. In this study, the liver volume is reduced in about half of the subjects, but how to consider the involvement of spontaneous reduction should be discussed.

RESPONSE: Thank you for pointing out this spontaneous effect. We totally agree that, in our study, the reduction of the annual growth rate of TKV was partially explained by this phenomenon because the patients of this study were older than previous studies. However, the logistic regression analysis adjusting related variables showed that higher growth rate of TLV as well as older age significantly predicted responders. Therefore, we consider that the reduction observed in our study would be not fully derived from the spontaneous reduction of PLD. 

Although it was difficult to estimate the rate of spontaneous reduction, we added this problem from page 24 line 12 to page 25 line 1.

Minor

Comment #1

In a stratified analysis of responder and non-responder, adjustments for variables have been described, however, specific method for this statistical procedure has not been described in the text.

RESPONSE: We wrote down the method on Page 10 line 14-18. 

Comment #2

Inconsistencies are seen in figure description in the text and figure numbering (e.g. Fig. 2B in P14L7). In addition, both Fig. 2 and Fig. 4 are not represented in A and B.

RESPONSE: Thank you for pointing out incorrect numbers of figure 2 and figure4. However, we decided to decline these figures in revised manuscript. 

Reviewer #4: 

RESPONSE: Thank you for providing us the suggestive comment on our article. Dose effect of tolvaptan and menopausal status were evaluated by conducting subgroup analysis. 

In addition, we set the control group after screening 902 patients again secondary analysis was conducted for patients without the history of intervention to polycystic liver to minimized the carry-over effect of such therapies. 

After reconsidering the values of figures, we decided to decline figure 2 and figure 3 because we considered that these figures did not directly represent the main result of this study. 

We hope that these revisions would make more appropriate this study design. 

Polycystic liver disease leads to pain and suffering in many individuals with autosomal dominant polycystic kidney disease. This interesting study examines changes in total liver volume in a cohort of individuals who took tolvaptan and followed longitudinally. About 50% experienced a reduction in liver volume and about 50% experienced a continued increase in liver volume. The paper is well written. However I am concerned about the statistics.

There are number of extreme outlier cases who could have influenced the data analysis. Overall we may be seeing a regression to the mean.

Please mark individuals that had TAE or liver volume reducing procedures on the graph.

RESPONSE: Thank you for the carry-over effect of the physical intervention of polycystic liver, such as liver trans-arterial embolization and liver cyst drainage. We decided to decline to use figure 2 and figure 4 in the revised manuscript however, we conducted the secondary analysis about patients without the history of physical intervention in this major revision. 

We added this subgroup analysis from page 21, line2 to page 22, line 8 and on Supporting Information. 

Please add figure showing both females that were pre and post menopausal and describe what happened to their total liver volumes.

RESPONSE: 

We collected the data about menopausal status about female patients in the tolvaptan group. Among them, we found the status only from 44 female patients. From the subgroup analysis for the groups according to the menopausal status, however, the annual growth rate of TLV was not significantly different before and after tolvaptan use. 

We added the result in Supporting Information and on page17 line 8-10.

If there is a positive effect of tolvaptan on total liver volume then one might expect a dose -response effect. No dose information has been provided.

RESPONSE: We conducted the subgroup analysis to evaluate the dose effect of tolvaptan. The tolvaptan group were divided into the four categories according to the maintenance dose of tolvaptan, however, there was no trend that suggested the dose effect of tolvaptan on response rate or decline of liver growth rate. We added the result of this subgroup analysis on page 17 line 8 and on Supporting Information Figure S1. 

Again, thank you for giving us the valuable comments in order to strengthen our manuscript. For these seven weeks we have been trying hard to incorporate your feedback and we hope that these revisions persuade you to consider the acceptance of our submission. 

Sincerely, 

Hiroki Mizuno.

Nephrology Center

Toranomon Hospital Kajigaya, Japan

hilomiz@yahoo.co.jp

+08-44-3588-5111

---

## [Decision Letter · Decision Letter 1]

25 Oct 2021

PONE-D-21-12669R1Potential effect of tolvaptan on polycystic liver disease for patients with ADPKD meeting the Japanese criteria of tolvaptan usePLOS ONE

Dear Dr. Mizuno,

Thank you for submitting your manuscript to PLOS ONE. After careful consideration, we feel that it has merit but does not fully meet PLOS ONE’s publication criteria as it currently stands. Therefore, we invite you to submit a revised version of the manuscript that addresses the points raised during the review process.

Most of the concerns are clarified, however, there still remains serious problem.  Authors should reanalyze and solve problem pointed out by a reviewer.

We look forward to receiving your revised manuscript.

Kind regards,

Tatsuo Shimosawa, M.D., Ph.D.

Academic Editor

PLOS ONE

Reviewers' comments:

Reviewer's Responses to Questions

**Comments to the Author**

1. If the authors have adequately addressed your comments raised in a previous round of review and you feel that this manuscript is now acceptable for publication, you may indicate that here to bypass the “Comments to the Author” section, enter your conflict of interest statement in the “Confidential to Editor” section, and submit your "Accept" recommendation.

Reviewer #2: All comments have been addressed

Reviewer #3: All comments have been addressed

Reviewer #4: All comments have been addressed

2. Is the manuscript technically sound, and do the data support the conclusions?

Reviewer #2: Yes

Reviewer #3: Partly

Reviewer #4: Partly

3. Has the statistical analysis been performed appropriately and rigorously? 

Reviewer #2: Yes

Reviewer #3: Yes

Reviewer #4: No

4. Have the authors made all data underlying the findings in their manuscript fully available?

Reviewer #2: Yes

Reviewer #3: Yes

Reviewer #4: No

5. Is the manuscript presented in an intelligible fashion and written in standard English?

Reviewer #2: Yes

Reviewer #3: Yes

Reviewer #4: No

6. Review Comments to the Author

Reviewer #2: The reviewer would like to thank the authors for revising the manuscript according to the comments. This

reviewer is satisfied with the revision.

Reviewer #3: Dear authors,

Thank you for responding to your opinion about the title. I believe that modification of the title has led to a more accurate representation of the study design. And, it is grateful to respond to the previous commentary that the case which carried out physical intervention should be analyzed as a subgroup.

In this revised version, total liver volume of patients administered tolvaptan was compared with that of patients who met the criteria of tolvaptan use in Japan and were not administered tolvaptan. However, this study has been conducted as a retrospective longitudinal observational study and is not a prospective randomized controlled trial.

Therefore, it is unlikely that patients who do not receive tolvaptan in this analysis are adequate as a control group because the selection bias of whether or not to use tolvaptan occurs when the treatment of choice was performed on the tolvaptan use. In addition, it is considered that it should be corrected to "non-tolvaptan" rather than “control”. However, it is favorable that the authors are trying to objectively assess the effectiveness of tolvaptan while there is a limitation in the retrospective study.

However, it is problematic that the authors described "53.6% of ADPKD patients taking tolvaptan experienced the reduction of the annual growth rate of TLV" as a conclusion. The outcome of this revised version is that tolvaptan is not efficacious in reducing hepatic cysts. Therefore, it may be concluded from statistical analyses that the reduction in TLV seen in about half of tolvaptan users is not of particular significance. I would like to know the author’s opinion on this issue.

The following is my personal opinions, but it is generally known that the TKV-reducing or-suppressing effects of tolvaptan are deviated in individual case. The same situation may be true for TLV reduction, and there may be an analysis of TLV reduction limitedly in the subgroup showing responder on TKV.

Minor

Are the ∆TLV% described on the Y-axis in Fig. 2 and annual liver growth rate on the X-axis in Fig. 4 not identical? If the same, the representations should be unified.

Reviewer #4: I still find that representation of the data as it is presented in the manuscript is problematic although The authors have striven to accommodate my concerns. There were more cyst drainage and HAEs in the Tolvaptan group than the control group. The manuscript body should focus on the tolvapan and control group differences excluding the patents who had other volume reductive interventions, specifically cyst drainage and hepatic artery embolization so as that it represents impact of Tolvaptan on PKD and not the multiple volume reducing procedures.

The manuscript body should report the study findings after the exclusion of these cases with cyst drainage and HAE from both groups as they complicated/confound the analysis. The main result is shown in figure S3 representing the correct group of patients with PLD who did not have any other volume -reducing procedures. This analysis should be moved to the main manuscript as the main result. The current figure representing changes in TLV including those with other volume reducing procedures should be moved to supplemental data. Text in abstract, results section and discussion of the paper should be edited accordingly. Figure 1 flow diagram should be modified to should reflect that these cases with volumes reductive procedures were removed from both tolvaptan and control group analyses.

If all cyst drainage procedures are aggregated as opposed to being divided by indication (cyst infection versus routing drainage), one would expect there will be a difference between cases and controls. Please also summate these cases in tables 1 and 2 to determine whether they have statistical significance between the two groups. How much fluid was removed by these drainage procedures? Please add this volumes to tables 1 and 2.

The 16 with volume reducing procedures should be removed from the figure 4 the responder/ non responder main analysis in the main body of the paper.

How was post menopausal defined? Please describe this in the methods.

Page 14, line 13: Analysis of whole patients- change this as it is not good English

Supplemental figures have no labels as uploaded.

Figure S4- is this the correct title?

7. PLOS authors have the option to publish the peer review history of their article (what does this mean?). If published, this will include your full peer review and any attached files.

Reviewer #2: No

Reviewer #3: No

Reviewer #4: No

---

## [Author Response · Author response to Decision Letter 1]

5 Jan 2022

Dear Editor and Reviewers

Thank you for reviewing our manuscript “Potential effect of tolvaptan on polycystic liver disease for patients with ADPKD meeting the Japanese criteria of tolvaptan use [PONE-D-21-12669R1]” and offering valuable advices. We also appreciate the time and effort you have dedicated to providing insightful feedback on our paper.

We have addressed to your comments with point-by-point responses and revised the manuscript accordingly. We would like to send two copies, clean and marked versions and as well as the pdf file of four figures.

Response to Editor and Reviewers’ comments

To Editor

Dear Editor

Thank you for reviewing our first manuscript.

In order to answer the reviewer#4’s suggestion, we decided to amend the whole part of the method and result of the first revision, because the patients with a history of physical interventions needed to be excluded in the main analysis. As a result, in this second revision, the number of patients in the cohort reduced from 107 to 62, however, we expect that this amendment would make our study clearer to understand the true effect of tolvaptan on PLD without bothering the carry-out effect of physical interventions of PLD.

It would be appreciated if you could review the new manuscript of the second revision.

Reviewer #2

Dear Reviewers

Thank you for reviewing our first manuscript.

In order to minimize the carry-out effect of physical interventions of PLD, which was pointed out by another reviewer, we excluded the patients having the previous history of such procedures including cyst drainage and trans-arterial embolization for liver arteries. In this second revision, however, the conclusion of the results was consistent with the first revision although the number of patients reduced from 107 to 62.

It would be grateful that you could review our manuscript in this new study design.

To Reviewer #3

Dear Reviewer #3: 

Thank you for pointing out the valuable suggestion to strengthen our manuscript. As reviewer #4 pointed out, in this second revision, the inclusion criteria were modified in order to exclude the carry-out effect of the physical intervention of PLD. The number of eligible patients decreased from 107 to 62, however, we believe this revision would make our study more efficient to analyze the tolvaptan’s effect on PLD.

“Thank you for responding to your opinion about the title. I believe that modification of the title has led to a more accurate representation of the study design. And, it is grateful to respond to the previous commentary that the case which carried out physical intervention should be analyzed as a subgroup.

In this revised version, total liver volume of patients administered tolvaptan was compared with that of patients who met the criteria of tolvaptan use in Japan and were not administered tolvaptan. However, this study has been conducted as a retrospective longitudinal observational study and is not a prospective randomized controlled trial.

Therefore, it is unlikely that patients who do not receive tolvaptan in this analysis are adequate as a control group because the selection bias of whether or not to use tolvaptan occurs when the treatment of choice was performed on the tolvaptan use. In addition, it is considered that it should be corrected to "non-tolvaptan" rather than “control”. However, it is favorable that the authors are trying to objectively assess the effectiveness of tolvaptan while there is a limitation in the retrospective study.”

RESPONSE: We agree with your idea of describing “non-tolvaptan group” because tolvaptan was not intended to be prescribed for these patients. We have changed all the expression from “control group to “non-tolvaptan group”

“However, it is problematic that the authors described "53.6% of ADPKD patients taking tolvaptan experienced the reduction of the annual growth rate of TLV" as a conclusion. The outcome of this revised version is that tolvaptan is not efficacious in reducing hepatic cysts. Therefore, it may be concluded from statistical analyses that the reduction in TLV seen in about half of tolvaptan users is not of particular significance. I would like to know the author’s opinion on this issue.”

RESPONSE: As well as in the first revision, the primary analysis of the difference of annual growth rate of tolvaptan in this second revision was not significantly different between tolvaptan group and non-tolvaptan group. Therefore, we considered that the tolvaptan’s effect on the annual growth rate of TLV was not established in this retrospective observational study, and then, we decided to describe the conclusion as follows.

“the change in annual growth rate of TLV in ADPKD patients taking tolvaptan was not statistically different compared with that in ADPKD patients without taking tolvaptan. However, tolvaptan may have the potential to suppress the growth rate of TLV in some PLD patients due to ADPKD, especially in older patients or those that are rapid progressors of PLD.”

“The following is my personal opinions, but it is generally known that the TKV-reducing or-suppressing effects of tolvaptan are deviated in individual case. The same situation may be true for TLV reduction, and there may be an analysis of TLV reduction limitedly in the subgroup showing responder on TKV.”

RESPONSE: Thank you for giving us the valuable suggestions about the association between TKV responder and TLV responder. Although the dose effect of tolvaptan on TLV was not found in our study, and as you mentioned the individual variability of tolvaptan’s effect existed in our cohort, in this cohort of secondary revision, we found that the response rates of TLV in patients with responder on TKV (64.9%) was larger than that in non-responder on TKV (45.2 %) (p<0.01). In addition, the change of annual liver growth rate was mildly associated with the change of the annual kidney growth rate (p<0.01) although rho was relatively lower level (rho=0.34). These facts may imply that some molecular mechanism expressed in both liver and kidney, such as polycystin-1 might be related to the effect of tolvaptan on liver or kidney although the expression rate of polycystin-1 was not detected in our study.

Minor

“Are the ∆TLV% described on the Y-axis in Fig. 2 and annual liver growth rate on the X-axis in Fig. 4 not identical? If the same, the representations should be unified.”

RESPONSE: delta TLV% represents the difference of annual liver growth rate between observational period and tolvaptan period, while TLV% in Fig 4 represents the annual liver growth rate. We unified these expressions between figures.

Reviewer #4: 

Dear Reviewer #4

Thank you for reviewing the manuscript of our first revision and giving us valuable suggestions to improve our study-design. We decided to adopt the new inclusion criteria you suggested. In order to answer all the comments you gave us, we had changed the method, result, and discussion part as well as figures and tables.

“I still find that representation of the data as it is presented in the manuscript is problematic although The authors have striven to accommodate my concerns. There were more cyst drainage and HAEs in the Tolvaptan group than the control group. The manuscript body should focus on the tolvapan and control group differences excluding the patents who had other volume reductive interventions, specifically cyst drainage and hepatic artery embolization so as that it represents impact of Tolvaptan on PKD and not the multiple volume reducing procedures.

The manuscript body should report the study findings after the exclusion of these cases with cyst drainage and HAE from both groups as they complicated/confound the analysis. The main result is shown in figure S3 representing the correct group of patients with PLD who did not have any other volume -reducing procedures. This analysis should be moved to the main manuscript as the main result. The current figure representing changes in TLV including those with other volume reducing procedures should be moved to supplemental data. Text in abstract, results section and discussion of the paper should be edited accordingly. Figure 1 flow diagram should be modified to should reflect that these cases with volumes reductive procedures were removed from both tolvaptan and control group analyses.”

RESPONSE:

As you suggested, the inclusion criteria were modified to exclude the physical interventions for PLD and we have changed the method section, the flow diagram of figure1 according to the new inclusion criteria. In addition, we have changed the main result of figure 2 and 3 by using the sub-group analysis of the manuscript of first revision by using figure S3 and S4.

“If all cyst drainage procedures are aggregated as opposed to being divided by indication (cyst infection versus routing drainage), one would expect there will be a difference between cases and controls. Please also summate these cases in tables 1 and 2 to determine whether they have statistical significance between the two groups. How much fluid was removed by these drainage procedures? Please add this volumes to tables 1 and 2.”

RESPONSE: After we have collected the data of cyst drainage, we found that there was no statistical difference of the volume of cyst drainage between tolvaptan group and the non-tolvaptan group. These data were added in table S1 and figure S2.

“The 16 with volume reducing procedures should be removed from the figure 4 the responder/ non responder main analysis in the main body of the paper.”

RESPONSE: We have excluded the 16 patients from figure 4.

“How was post menopausal defined? Please describe this in the methods.”

RESPONSE: The menopausal status was determined by self-reported questionnaire. We added the method on page 8 line 8.

“Page 14, line 13: Analysis of whole patients- change this as it is not good English”

RESPONSE: We have changed the expression as following

“Analysis of the patients including the history of physical intervention for the polycystic liver

”

“Supplemental figures have no labels as uploaded.”

RESPONSE: We had added the labels of supplemental figures.

“Figure S4- is this the correct title?”

RESPONSE: We have corrected the titles of figure S3 and S4.

Again, thank you for giving us the valuable comments in order to strengthen our manuscript. We hope that these revisions persuade you to consider the acceptance of our submission.

Sincerely,

Hiroki Mizuno.

Nephrology Center

Toranomon Hospital Kajigaya, Japan

hilomiz@yahoo.co.jp

+08-44-3588-5111

---

## [Decision Letter · Decision Letter 2]

3 Feb 2022

Potential effect of tolvaptan on polycystic liver disease for patients with ADPKD meeting the Japanese criteria of tolvaptan use

PONE-D-21-12669R2

Dear Dr. Mizuno,

We’re pleased to inform you that your manuscript has been judged scientifically suitable for publication and will be formally accepted for publication once it meets all outstanding technical requirements.

Kind regards,

Tatsuo Shimosawa, M.D., Ph.D.

Academic Editor

PLOS ONE

Additional Editor Comments (optional):

Reviewers' comments:

Reviewer's Responses to Questions

**Comments to the Author**

1. If the authors have adequately addressed your comments raised in a previous round of review and you feel that this manuscript is now acceptable for publication, you may indicate that here to bypass the “Comments to the Author” section, enter your conflict of interest statement in the “Confidential to Editor” section, and submit your "Accept" recommendation.

Reviewer #3: (No Response)

Reviewer #4: All comments have been addressed

2. Is the manuscript technically sound, and do the data support the conclusions?

Reviewer #3: Yes

Reviewer #4: Yes

3. Has the statistical analysis been performed appropriately and rigorously? 

Reviewer #3: Yes

Reviewer #4: Yes

4. Have the authors made all data underlying the findings in their manuscript fully available?

Reviewer #3: Yes

Reviewer #4: Yes

5. Is the manuscript presented in an intelligible fashion and written in standard English?

Reviewer #3: Yes

Reviewer #4: Yes

6. Review Comments to the Author

Reviewer #3: I thank you for your sincere responses. I accept all of your responses. Just one, “Control” in Table 1 should be replaced with “non-tolvaptan group”.

Reviewer #4: The authors have addressed my concerns in their rebuttal. The revised manuscript is much improved.

7. PLOS authors have the option to publish the peer review history of their article (what does this mean?). If published, this will include your full peer review and any attached files.

Reviewer #3: No

Reviewer #4: No

---

## [Editor Report · Acceptance letter]

7 Feb 2022

PONE-D-21-12669R2 

Potential effect of tolvaptan on polycystic liver disease for patients with ADPKD meeting the Japanese criteria of tolvaptan use 

Dear Dr. Mizuno:

I'm pleased to inform you that your manuscript has been deemed suitable for publication in PLOS ONE. Congratulations! Your manuscript is now with our production department. 

Kind regards, 

on behalf of

Prof. Tatsuo Shimosawa 

Academic Editor

PLOS ONE